# Detection of Bird Nests on Transmission Towers in Aerial Images Based on Improved YOLOv5s

**Gujing Han** [1,2,*], **Ruijie Wang** [1,2], **Qiwei Yuan** [1,2], **Saidian Li** [1,2], **Liu Zhao** [1,2], **Min He** [3], **Shiqi Yang** [3] **and Liang Qin** [3]

1  Department of Electronic and Electrical Engineering, Wuhan Textile University, Wuhan 430200, China
2  State Key Laboratory of New Textile Materials and Advanced Processing Technologies,
   Wuhan Textile University, Wuhan 430200, China
3  School of Electrical and Automation, Wuhan University, Wuhan 430072, China
*  Correspondence: gjhan@wtu.edu.cn; Tel.: +86-133-8754-5297

**Abstract:** To further improve the accuracy of bird nest model detection on transmission towers in aerial images without significantly increasing the model size and to make detection more suitable for edge-end applications, the lightweight model YOLOv5s is improved in this paper. First, the original backbone network is reconfigured using the OSA (One-Shot Aggregation) module in the VOVNet and the CBAM (Convolution Block Attention Module) is embedded into the feature extraction network, which improves the accuracy of the model for small target recognition. Then, the atrous rates and the number of atrous convolutions of the ASPP (Atrous Spatial Pyramid Pooling) module are reduced to effectively decrease the parameters of the ASPP. The ASPP is then embedded into the feature fusion network to enhance the detection of the targets in complex backgrounds, improving the model accuracy. The experiments show that the mAP (mean-Average Precision) of the fusion-improved YOLOv5s model improves from 91.84% to 95.18%, with only a 27.4% increase in model size. Finally, the improved YOLOv5s model is deployed into the Jeston Xavier NX, resulting in a model that runs well and has a substantial increase in accuracy and a speed of 10.2 FPS, which is only 0.7 FPS slower than the original YOLOv5s model.

**Keywords:** aerial images; bird nest detection; YOLOv5s; model deployment

## 1. Introduction

According to statistics, bird activities are the third most common cause of high-voltage transmission line faults, after lightning damage and external damage. Furthermore, bird activities on high-voltage transmission towers account for 90% of failures. As a result, an excellent method for detecting bird nests can better prevent electrical failures caused by bird activities [1–3].

As most transmission towers are located in fields, and bird nests are mostly built high up in such towers, traditional manual inspection is time-consuming and inefficient [4,5]. With the continuous development of artificial intelligence technology, UAV(Unmanned Aerial Vehicle) inspections are gradually replacing manual inspections. Many intelligent object detection algorithms based on image recognition techniques have been applied to bird nest detection [6,7].

There are two broad categories for detecting bird nests in aerial images in UAV inspections. One category consists of traditional transmission line object detection algorithms based on feature extraction and classification models [8–11]; however, such algorithms can lack robustness due to the irregular shapes of bird nests and the fact that their color changes as the light changes. The second category is object detection algorithms based on deep learning, including two-stage and one-stage algorithms. The two-stage detection algorithm is executed in two steps: first, the candidate box region is obtained and then

the classification of targets within the region is performed; representative algorithms include R-CNN (Rigion-based Convolution Neural Networks), Fast R-CNN, Mask R-CNN, Faster R-CNN, etc. [12–15]. In ref. [16], the authors proposed an algorithm for bird nest detection based on an improved Faster R-CNN. The effect of complex backgrounds on detecting bird nests was excluded by simultaneously detecting bird nests and transmission towers. The two-stage detection algorithm provides a significant improvement in accuracy over the traditional algorithm, but the detection process needs to be completed in two steps, and the detection speed is slow. The one-stage detection algorithm predicts the class and region of the target directly through the object detection network; representative algorithms include SSD (Single shot multibox detector) and YOLO (You Only Look Once) series [17–22]. In ref. [23], the authors replaced the backbone network of the SSD with EfficientNetB7, reducing the model parameters. In ref. [24], the authors combined SSD with HSV (Hue Saturation Value) spatial color filters to improve model accuracy; however, the SSD network did not have the FPN (Feature Pyramid Networks) structure, resulting in a lower accuracy than the YOLO series. As the representative algorithm in the YOLO series, YOLOv3 improved detection accuracy while maintaining the fast detection speed of the one-stage algorithm. In ref. [25], the authors replaced the standard convolution in the backbone of the YOLOv3 network with the depth-separable convolution, thus reducing the network parameters and improving the speed of the detection network; however, the detection of small targets was still inadequate. YOLOv4 builds on the strengths of YOLOv3 and further enhances detection accuracy. In ref. [26], the authors added the Swin Transformer module to the backbone of the YOLOv4 network to address the difficulty of multi-scale detection, but model parameters were too large to be deployed at the edge. In ref. [27], the authors introduced self-attention in the YOLOv4-Tiny network and ported the model to an embedded platform, but it was not accurate enough for target recognition in complex contexts. YOLOv5 has improved the accuracy of YOLOv4 while presenting a lighter version, YOLOv5s. In ref. [28], the authors removed a $20 \times 20$ prediction head from the prediction network part of YOLOv5s and added a $160 \times 160$ prediction head to improve the model accuracy, but it did not detect small targets effectively. In ref. [29], the authors replaced the feature fusion network in YOLOv5s with Bi-FPN (Bidirectional Feature Pyramid Network), which improved the model accuracy; however, the structure of Bi-FPN was too complex, leading to a dramatic increase in model size and reducing speed.

YOLOv5 offers higher detection results and detection speed than the YOLOv3 and YOLOv4 algorithms, and its lightweight version, YOLOv5s, achieves 73 FPS on the COCO dataset with a model size of only 27.8 M. However, there are still cases of false detections and misdetections for small targets and targets in complex contexts [30,31]. Complex improvements to the feature fusion layer make the model too large to be applied on embedded devices [32]. To address the above problems, this paper proposes a target detection algorithm for bird nests on transmission towers in aerial images based on improved YOLOv5s. Unlike conventional algorithms, the algorithm in this paper can significantly improve the accuracy without significantly increasing the model size and parameters, thus enabling its successful deployment in the Jetson Xavier NX. The specific aims of the work are as follows: (1) To address the problem of low accuracy of the model, the backbone network of the YOLOv5s model is reconstructed to improve the model accuracy; (2) To address the problem that small targets of distant bird nests are not easy to detect, the attention mechanism is embedded in the feature extraction network to enhance the detection of small targets; (3) To address the problem of poor bird nest detection in complex backgrounds, the ASPP module is improved to enhance the detection of targets in complex backgrounds without significantly increasing the model size.

## 2. Structure and Features of the YOLOv5 Model

The YOLOv5 target detection algorithm can be divided into four releases: YOLOv5s, YOLOv5m, YOLOv5l, and YOLOv5x, depending on the width and depth of the network. Considering the need to deploy the model to embedded devices at a later stage, the

lightweight YOLOv5s was chosen as the base model in this paper. The structure of the YOLOv5s object detection network can be divided into three parts: feature extraction network, feature fusion network, and prediction network, as shown in Figure 1.

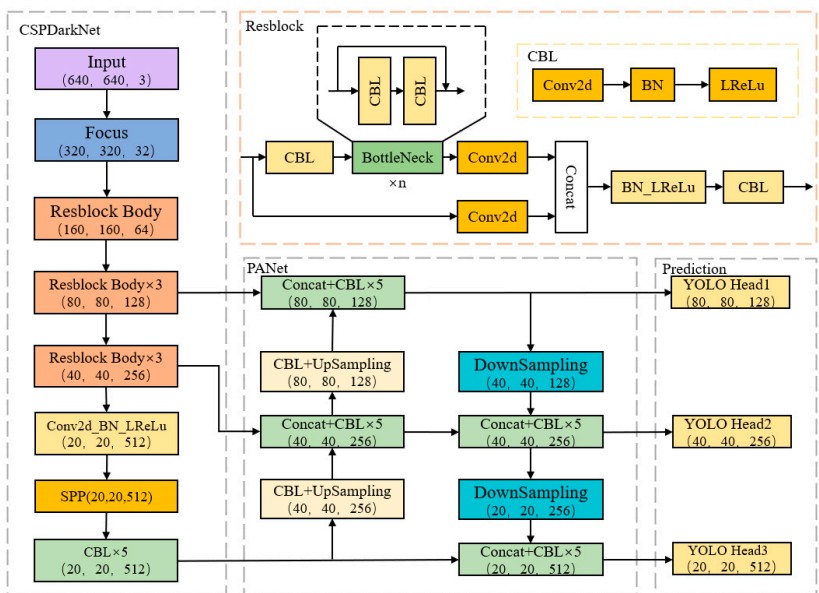

**Figure 1.** The structure of the original YOLOv5s.

The feature extraction network follows the CSPDarkNet in YOLOv4, splitting the residual blocks in DarkNet in the channel dimension direction of the feature map into two parts. Of these, a portion passes typically through the convolution and bottleneck layers, keeping with the structure of Resblock. The other part is spliced with the output of another branch in the direction of the channel of the feature map after a small amount of convolution. This approach reduces computational effort while avoiding problems such as gradient disappearance.

The feature fusion network uses the SPP (Spatial Pyramid Pooling) structure and the PANet (Path Aggregation-Network). The SPP structure solves the problem that the input image size of CNN (Convolutional Neural Networks) must be fixed and can make the input image size unrestricted; the SPP structure is shown in Figure 2.

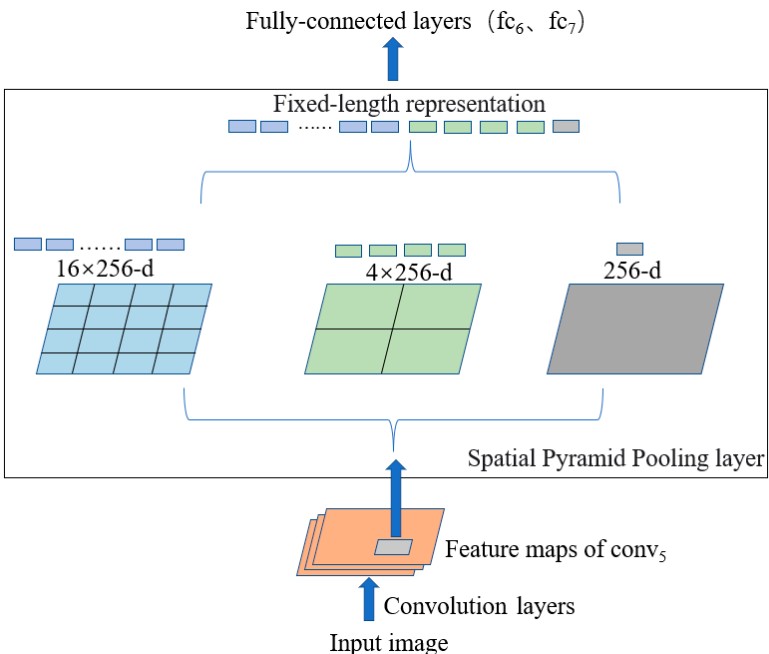

**Figure 2.** The structure of the SPP.

PANet uses the three feature layers of 80 × 80, 40 × 40, and 20 × 20 extracted by FPN to fuse bottom-up, which can obtain three fused features to improve the effect of target detection at different scales; the structure is shown in Figure 3.

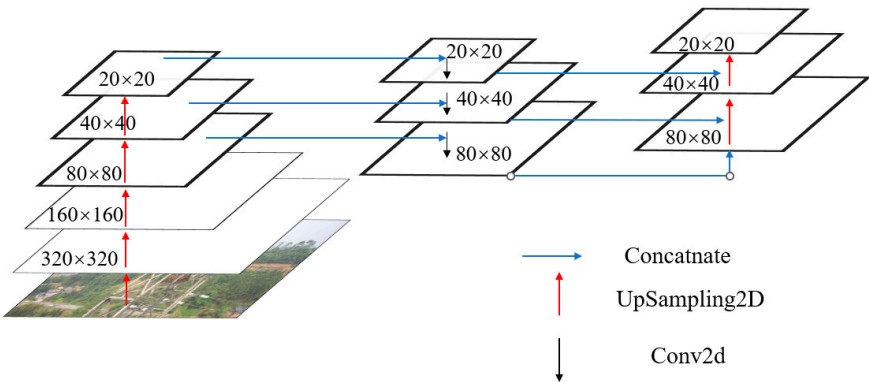

**Figure 3.** The structure of the PANet.

The prediction network implements the prediction of the final information, with three different scales of prediction heads corresponding to three different sizes of fused features. In the model prediction, the prediction network ranks the confidence levels of the N categories of the anchors from highest to lowest, and the threshold setting and redundant borders are removed using a non-maximum suppression algorithm to obtain the category and location information of the predicted target.

## 3. Recognition of Bird Nests on Transmission Towers Based on Improved YOLOv5s

### 3.1. Feature Extraction Network Based on OSA Module

To improve the model accuracy, this paper adopts the OSA Block in VOVNet to reconstruct the feature extraction network based on YOLOv5s, replacing the ResBlock in the original feature extraction network with OSA Block.

The OSA Block is an improvement on the Dense Block (whose structure is shown in Figure 4a) in the DenseNet [33,34]. It does away with the fusion method, where each layer aggregates the previous layer to avoid information redundancy. It takes the form of fusing

all the previous layers at once in the last layer, so that the input size remains the same and new output channels can be expanded while reducing the number of parameters. The structure of the OSA Block is shown in Figure 4b.

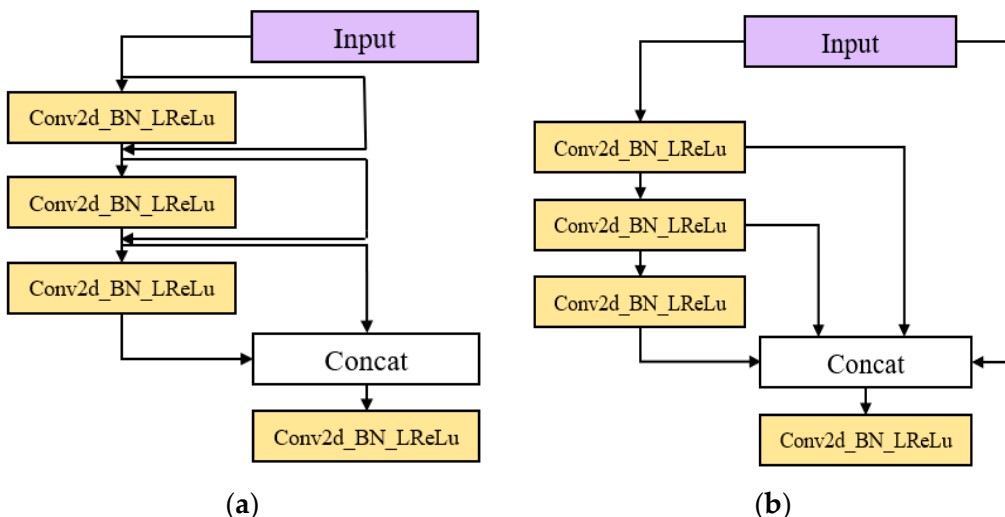

(**a**)       (**b**)

**Figure 4.** (**a**) Structure of the Dense Block; (**b**) structure of the OSA Block.

*3.2. The Impact of Attentional Mechanism Fusion Studies on the Effectiveness of Small Target Detection*

As the feature extraction network continues to deepen and the information extracted at the output becomes progressively more abstract, the initial model is not good enough for small target detection. This paper addresses this problem by incorporating attention mechanisms into the network. The attention mechanism refers to the human behavior of selectively focusing on the important parts of the received information to construct a model that can redistribute the weight of the target information from the irrelevant information in the information received by the network. The SAM (spatial attention module) is primarily designed to capture the correlation between pixel points in the input features; the CAM (channel attention module) aims to enhance the feature channels and amplify the target weights, improving the detection of small targets [35–37].

This paper chooses the CBAM which combines the SAM with the CAM [38]. It consists of CAM and SAM sub-modules. It can perform separate attention operations in the channel and spatial dimensions, reducing the model parameters, as shown in Figure 5.

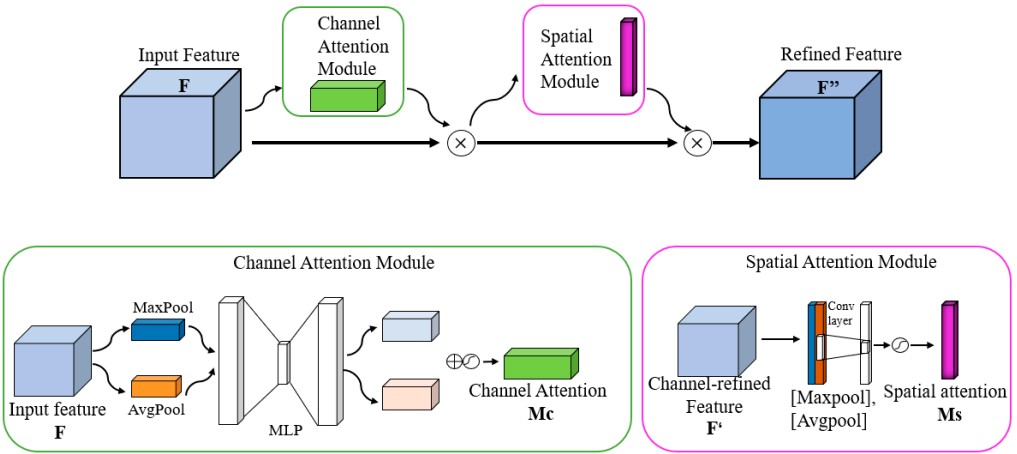

**Figure 5.** The structure of the CBAM.

Assuming that the input features are *F* and the one-dimensional channel convolution in the CAM is $M_c$, then the principle of the CAM is as in Equation (1). Using the CAM

output $F'$ as the input to the SAM. Assume that the two-dimensional spatial convolution in the spatial attention module is $M_s$. The principle of the spatial attention module is then given by Equation (2).

$$M_s(F) = \sigma(MLP(AvgPool(F)) + MLP(MaxPool(F))) \tag{1}$$

$$M_s(F) = \sigma\left(f^{7\times7}([AvgPool(F); MaxPool(F)])\right) \tag{2}$$

In Equations (1) and (2), $\sigma$ is a sigmoid function and $f^{7\times7}$ is a convolution kernel of size $7 \times 7$.

The total process of the operation of the CBAM module is shown in Equations (3) and (4).

$$F' = M_c(F) \otimes F \tag{3}$$

$$F'' = Ms(F') \otimes F' \tag{4}$$

The attention mechanism is already widely used in some networks due to its plug-and-play convenience. However, there is no definitive answer as to which part of the network it is better to embed it in. In this paper, two different embedding methods at different locations are designed. One is to integrate the CBAM after the Concat layer in the OSA module, which is in the feature extraction network; the other is to integrate it after the Concat layer in the feature fusion network.

The first embedding method is shown in Figure 6a. We embed the CBAM after the Concat layer of the OSA module. Firstly, the CBAM reconfigures the network structure through the CAM to assign more weight to important semantic information. Secondly, it performs the SAM to compress the number of channels and give better accuracy and a lower error rate.

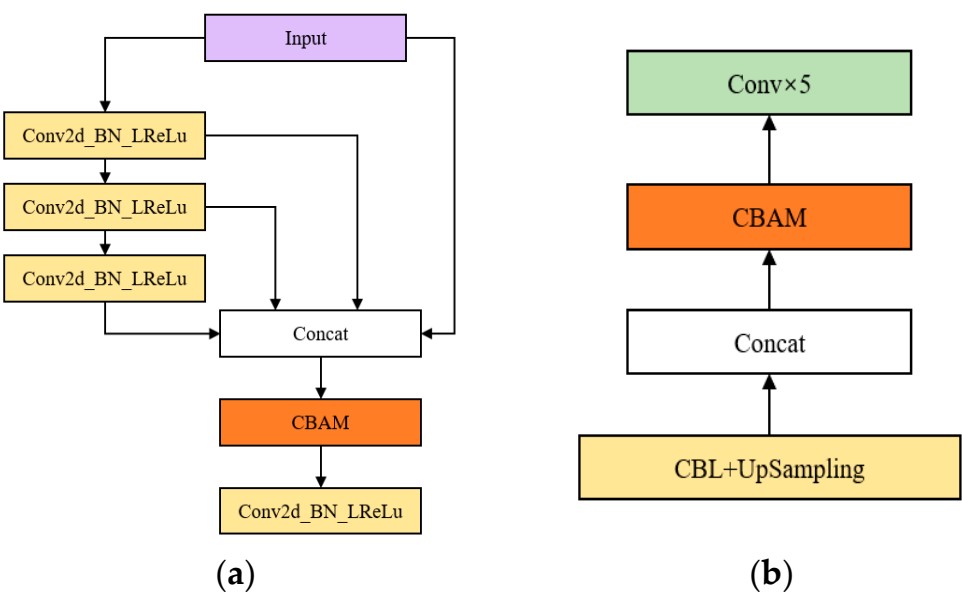

**(a)**　　　　　　　　　　　　**(b)**

**Figure 6.** (**a**) CBAM fusion in feature extraction network; (**b**) CBAM fusion in feature fusion network.

The second embedding method is shown in Figure 6b. We embed CBAM after the Concat layer in the feature fusion network. The PAN and FPN structures in the feature fusion network allow top-down delivery of semantic information and bottom-up delivery of localization information, and the fusion of deep and shallow information is achieved through four Concat layers. It is possible to assign more weight to the fused feature information if the CBAM module is placed behind the Concat layer.

### 3.3. Modified ASPP to Enhance Targets Detection in Complex Background

Bird nests on high-voltage transmission towers are mostly in complex contexts, which makes detection much more difficult. In this paper, we introduce the ASPP module to sample the input feature maps according to an atrous convolution of different atrous rates and then fuse the obtained results for average pooling to improve target recognition in complex backgrounds. The structure of the ASPP is shown in Figure 7a. Traditional downsampling increases the field of perception but reduces spatial resolution. Using atrous convolution guarantees resolution while expanding the field of perception. However, the atrous convolution is computed in a chessboard-like format, in which the results obtained at one layer, from an independent set of the previous layer, are not interdependent. If you increase the atrous rate to increase the field of perception of the model, you will lose core information and make the model less accurate. Therefore, this paper improves on the original ASPP structure by turning the four-branch atrous convolution into the three-branch atrous convolution, reducing the number of model parameters. We change the atrous rates from (6, 12, 18, 24) to (3, 5, 7); although this does not enhance the perceptual field as effectively as the original, it avoids information loss as much as possible and thus improves accuracy. The improved structure of the ASPP is shown in Figure 7b.

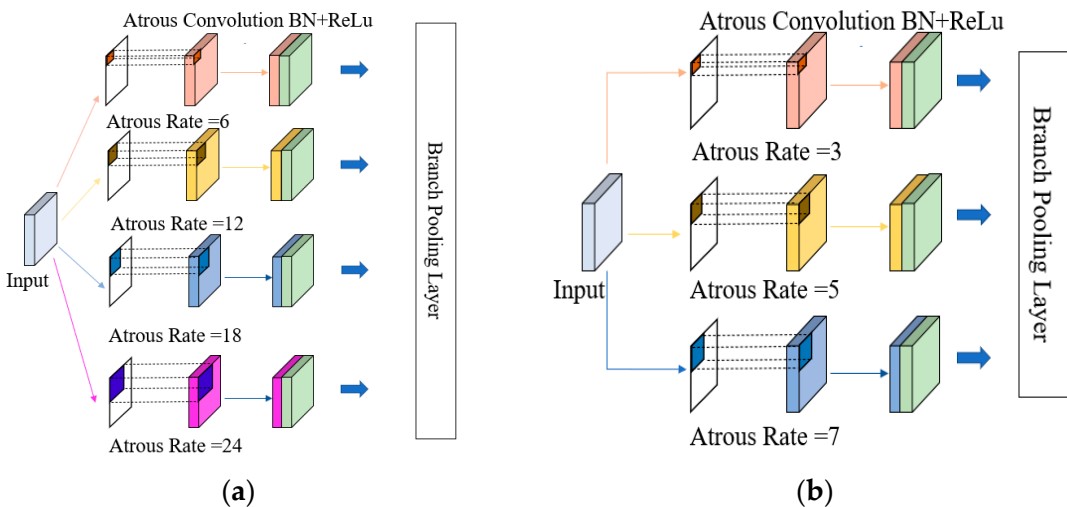

**Figure 7.** (**a**) Structure of the original ASPP; (**b**) structure of the improved ASPP.

### 3.4. The Improved YOLOv5 Algorithm Structure

In summary, the structure of the improved YOLOv5 algorithm is shown in Figure 8. After the input images pass through the feature extraction network, the OSA Block lets the information from the upper to lower layers intersect without losing the core information. After the information intersection, CBAM recalibrates the target feature information and amplifies the target weights to obtain three effective feature layers of different sizes, enhancing the detection of small targets. The feature fusion network is improved with the ASPP module to enhance the perceptual field of each input feature layer. Combining information from different scale feature layers provides the basis for improving the detection accuracy of the model for targets in complex backgrounds.

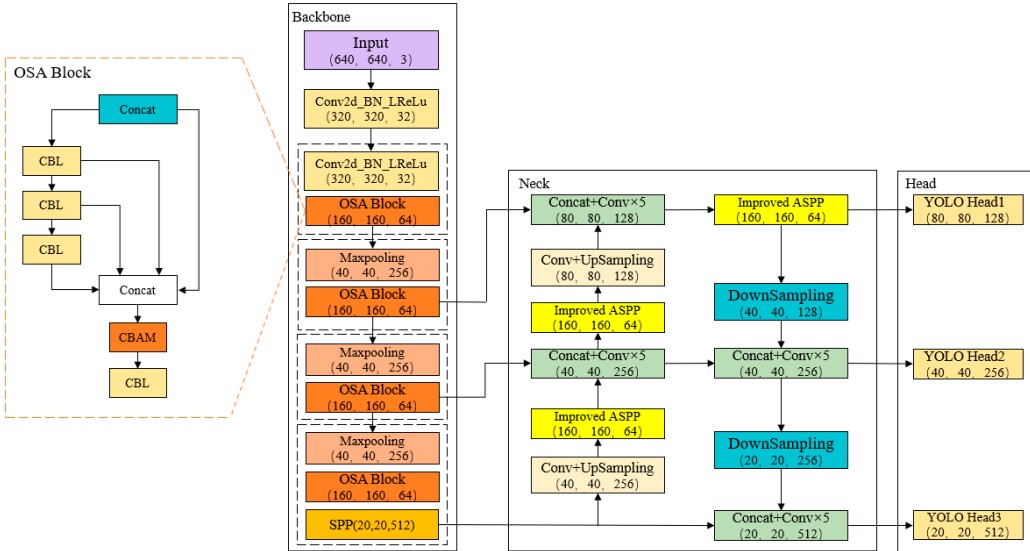

**Figure 8.** The structure of the improved YOLOv5s.

## 4. Experimental Results and Analysis

### 4.1. Data Structure and Processing

In this paper, aerial images of bird nests on high-voltage transmission towers are used as the object of study, and the experimental data are sourced from a power company. In this experiment, 1720 original images are expanded to 2644 images, including a total of 3759 target objects, using data augmentation methods such as flip, light and dark transformation, and Gaussian noise. The ratio of the training set, validation set, and test set is 8:1:1. The final result was 2116 training sets, 264 validation sets, and 264 test sets, and each image was annotated by hand using the LabelImg annotation tool. Figure 9 shows several representative aerial UAVs images. In Figure 9, the bird nest targets are framed in red.

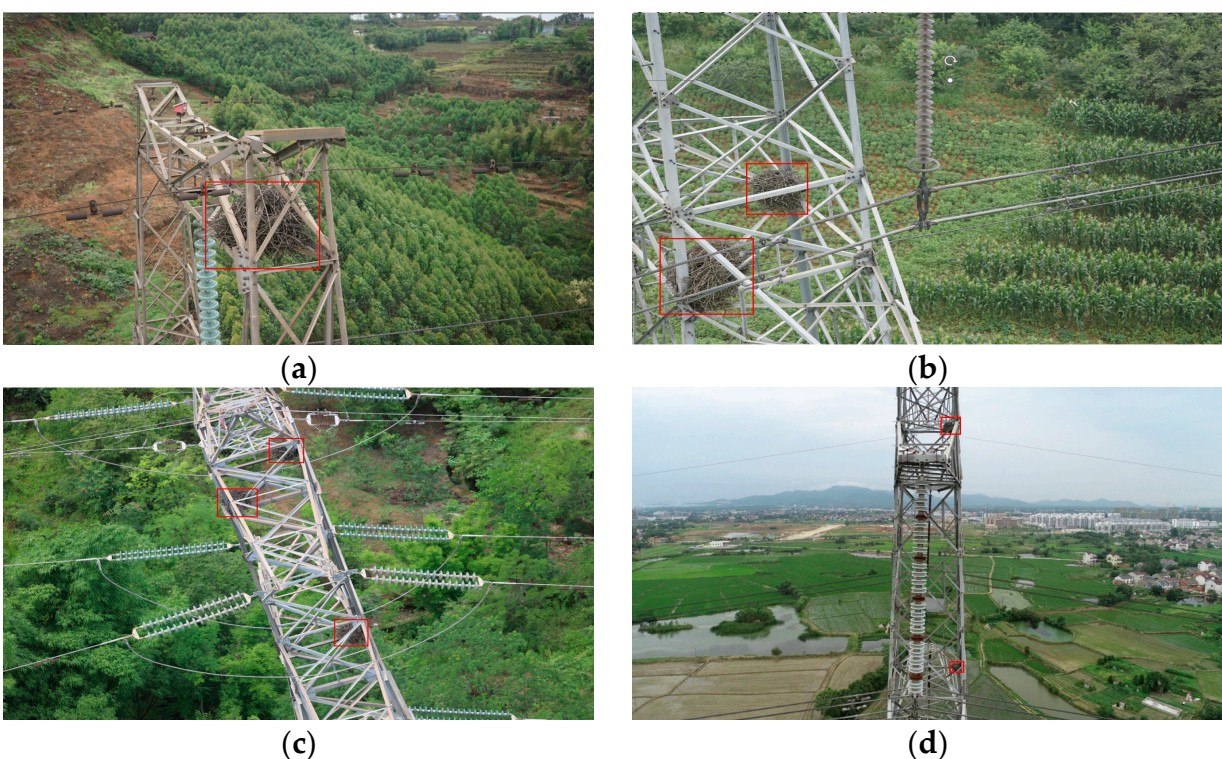

**Figure 9.** (**a**) Single-target bird nest; (**b**) multi-target bird nests; (**c**) bird nest targets in complex backgrounds; (**d**) bird nests as small targets.

### 4.2. Experimental Environment

This paper uses a deep learning framework based on the PyTorch 1.6.0 environment with Ubuntu 20.04, python 3.6.10, and CUDA = 11.4, where the training graphics card configuration is an NVIDIA RTXA6000/48G graphics card. The local computer NVIDIA GeForce RTX 3060Ti 8G is used for the trained model test.

### 4.3. Training Process

During model training, the backbone structure is changed so that no pre-training weights are used for training. To reduce the likelihood of the model falling into a local optimum, the SGD (Stochastic Gradient Descent) optimizer is used. It is trained with 300 epoch rounds and a batch size of 16. The cosine annealing learning rate is used to decay the learning rate of the bias layer to improve the convergence speed of the model.

### 4.4. Evaluation Indexes

This paper uses common performance evaluation indicators for target detection to compare different algorithms, i.e., Precision, Recall, mAP, FPS (Frames Per Second), Param Size, Total Params, FNR (False Negative Ratio), and FDR (False Detection Ratio).

Precision is shown in Equation (5), where TP (True Positive) represents positive samples predicted by the model to be in a positive class, and FP (False Positive) represents negative samples predicted by the model to be in a positive class. Precision represents the ratio of the actual positive samples to all the positive samples predicted by the model.

$$\text{Precision} = \frac{\text{TP}}{\text{TP} + \text{FP}} \tag{5}$$

Recall is shown in Equation (6), where FN(False Negative) represents the positive samples predicted by the model to be in a negative class. Recall represents the ratio of positive samples predicted by the model to actual positive samples [39].

$$\text{Recall} = \frac{TP}{TP + FN}$$

(6)

$$S = \int_0^1 p(r)\mathrm{d}r$$

(7)

The P–R curve plotted with Recall as the horizontal axis and Precision as the vertical axis is integrated to find the area under the curve as mAP, denoted as *S*, as shown in Equation (7). The P–R curve for the improved model in this paper is shown in Figure 10.

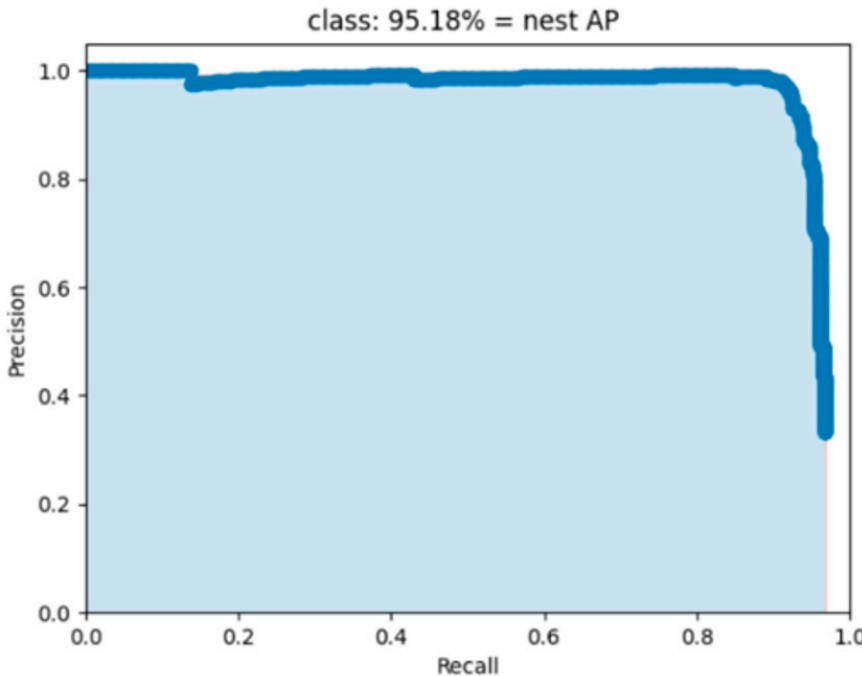

**Figure 10.** The P–R curves for the improved model.

In this paper, we introduced FNR as an index to evaluate the effectiveness of small target detection before and after model improvement. FNR is shown in Equation (8), which represents the ratio of positive samples predicted by the model to be negative samples to all positive samples present.

Furthermore, we introduced FDR as an index to evaluate the effectiveness of the target detection in complex backgrounds before and after model improvement. FDR is shown in Equation (9), which represents the ratio of negative samples predicted as positive by the model to all samples predicted as positive by the model.

$$\text{FNR} = \frac{FN}{FN + TP}$$

(8)

$$\text{FDR} = \frac{FP}{FP + TP}$$

(9)

*4.5. Comparison of Experimental Results*

4.5.1. Cross-Direction Comparison of Experimental Results

Table 1 shows the test results of the current mainstream deep learning algorithms on the dataset of this paper [40,41].

**Table 1.** Horizontal comparison of different algorithms.

| Models | mAP/% | Speed/FPS | Model Size/MB |
|---|---|---|---|
| Faster R-CNN | 89.65 | 14 | 113 |
| SSD | 90.48 | 101 | 100.27 |
| YOLOv3 | 89.10 | 55 | 236 |
| YOLOv4 | 90.35 | 44 | 245 |
| YOLOv4-Tiny | 80.12 | 175 | 23 |
| Original YOLOv5s | 91.84 | 73 | 27.8 |
| YOLOv5m | 93.77 | 51 | 81.54 |
| YOLOXs | 94.52 | 76 | 35.9 |
| YOLOv6s | 93.9 | 65 | 17.2 |
| YOLOv7-Tiny | 91.96 | 88 | 24.2 |
| Improved YOLOv5s | 95.18 | 55 | 37.85 |

The following four conclusions can be drawn from Table 1:

1. Among the five mainstream algorithms of Faster R-CNN, SSD, YOLOv3, YOLOv4, and YOLOv4-Tiny, since Faster R-CNN is a two-stage algorithm, it has the highest mAP at 90.48% but the slowest speed is only 10.1FPS, less than 14% of YOLOv5s speed. YOLOv4-Tiny, with only two prediction heads, has the fastest model detection speed at 175 FPS but the lowest mAP at 80.12, which sacrifices detection accuracy for increased detection speed;

2. The base model chosen in this paper, YOLOv5s, has higher accuracy than all the previous five algorithms and is second only to YOLOv4-Tiny in terms of detection speed and model size, which is why it is chosen as the base algorithm in this paper;

3. The fusion improvement algorithm in this paper improves by 3.34% over the original YOLOv5s, with only a 27.6% increase in model size. Compared to YOLOv5m, which improves network depth and width in YOLOv5, the mAP is improved by 1.41%, the model size is only 46.4% of YOLOv5m, and the detection speed is 4 FPS higher than YOLOv5m, which is eligible for deployment in embedded devices;

4. As Figure 11 shows, the improved fusion algorithm in this paper outperforms the mAP of the latest YOLO series of algorithms such as YOLOXs, YOLOv6s, and YOLOv7-Tiny. Moreover, there is a little difference in terms of speed. This shows that the improved algorithm in this paper is still competitive even when compared to the latest algorithms.

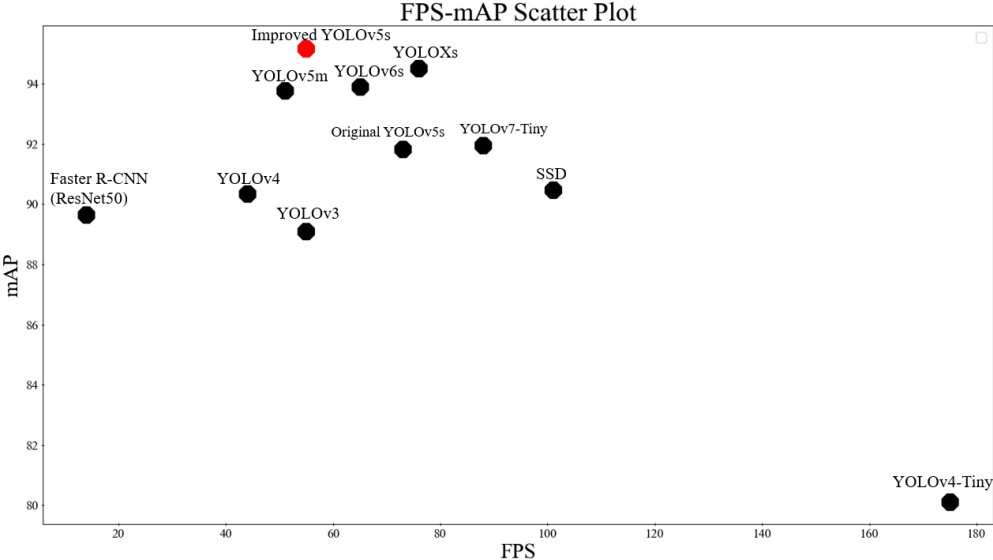

**Figure 11.** The FPS–mAP scatter plots.

### 4.5.2. Longitudinal Comparison of Experimental Results

Table 2 shows the algorithms corresponding to the first two improvements in this paper (YOLOv5s-V, YOLOv5s-V-att1, YOLOv5s-V-att2, in that order).

**Table 2.** Test results for the fusion of reconstructed backbone and attention mechanisms.

| Models | mAP/% | Speed/FPS | FNR/% | Model Size/MB |
|---|---|---|---|---|
| YOLOv5s | 91.84 | 73 | 2.84 | 27.8 |
| YOLOv5s-V | 93.08 | 65 | 2.46 | 30.2 |
| YOLOv5s-V-att1 | 93.37 | 61 | 2.11 | 31.28 |
| YOLOv5s-V-att2 | 94.41 | 61 | 1.76 | 31.58 |

Combining the analysis in Table 2 and Figure 12, the following four conclusions can be drawn:

1. The YOLOv5s-V algorithm only reconstructs the backbone network with the OSA Block in VOVNet. The mAP is increased from 91.84% to 93.08%; FNR is reduced by 0.38%; and the model size is increased by only 2.4 MB;

2. The YOLOv5s-V-att1 algorithm is based on YOLOv5s-V, after adding the CBAM attention mechanism to the Concat layer in the feature fusion network. YOLOv5s-V-att1 has a 0.29% increase in mAP compared to YOLOv5s-V without the attention mechanism module, which is not a significant increase. This is because, for feature fusion networks, where the CBAM attention mechanism is added after the feature fusion Concat layer after ResBlock, the feature extraction network loses some of the semantic information;

3. The YOLOv5s-V-att2 algorithm is based on YOLOv5s-V, after adding the CBAM attention mechanism to the Concat layer in the OSA module of the feature extraction network. YOLOv5s-V-att2 has a 1.33% improvement in mAP compared to the original YOLOv5s-V. For the feature extraction network, CBAM performs spatial and channel attention on the fused Concat layer of features in the OSA module, which is good for information retention and weight assignment. Therefore, this paper chose to use the YOLOv5s-V-att2 algorithm, which adds an attention mechanism to the feature extraction network;

4. Compared to the original YOLOv5s, the improved backbone YOLOv5s-V and the YOLOv5s-V-att2 with the attention mechanism have both improved on the aspects of FNR. YOLOv5s-V-att2 improved by 1.08% compared to the original model, which indicates that the improved model has a good improvement in small target detection.

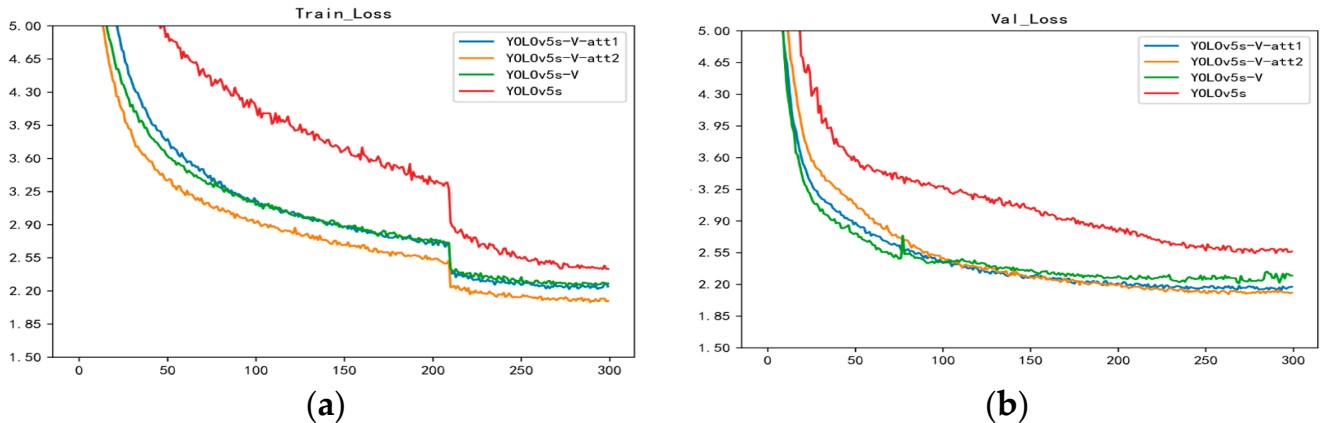

**Figure 12.** (**a**) Training loss curves for the models are shown in Table 2; (**b**) validation loss curves for the models are shown in Table 2.

### 4.5.3. Experimental Comparison of the Improved ASPP Module

Table 3 shows the test results on the original YOLOv5s model with the ASPP module set to the different numbers of atrous convolutions and atrous rates.

**Table 3.** ASPP module improvement test experiment.

| Experiments | The Number of Atrous Convolutions | Atrous Rates | mAP/% | Model Size/MB |
|---|---|---|---|---|
| I | 4 | (6, 12, 18, 24) | 92.95 | 38.11 |
| II | 3 | (6, 12, 18) | 91.73 | 34.35 |
| III | 5 | (6, 12, 18, 24, 30) | 92.58 | 41.86 |
| IV | 3 | (4, 6, 8) | 92.92 | 34.35 |
| V | 3 | (3, 5, 7) | 93.01 | 34.35 |
| VI | 3 | (2, 4, 6) | 92.65 | 34.35 |
| VII | 4 | (3, 5, 7, 9) | 93.28 | 38.11 |

The following three conclusions can be drawn from Table 3:

1. From Experiments I, II, and IV, we find that the number of atrous convolutions affects the model parameters but has little effect on mAP; the atrous rate has a greater effect on mAP;
2. From experiments II, IV, V, and VI, we find that the ASPP module with the atrous rate of (3, 5, 7) has the best result, with the mAP 0.06% higher than the original ASPP module with the atrous convolution number of 4 and the model size reduction of 3.76 MB;
3. From experiments V and VII, we find that the mAP is increased by 0.27%, and the model size is increased by 3.76 MB by increasing the number of atrous convolutions by one proportional to the atrous rate along (3, 5, 7). Combining mAP and model size, the ASPP module with many atrous convolutions of 3 and the atrous rate of (3, 5, 7) is chosen in this paper.

### 4.5.4. Comparison of Results of Ablation Experiments

Table 4 shows the comparison of three improved algorithms for ablation experiments.

**Table 4.** Experimental results of ablation with three improved algorithms.

| Models | mAP/% | Speed/FPS | FDR/% | Model Size/MB |
|---|---|---|---|---|
| YOLOv5s | 91.84 | 73 | 6.56 | 27.8 |
| YOLOv5s-V | 93.08 | 65 | 6.44 | 30.2 |
| YOLOv5s-V-att2 | 94.41 | 61 | 5.9 | 31.58 |
| YOLOv5s-V-att2(No SPP) | 93.86 | 63 | 6.4 | 29.58 |
| YOLOvs5-V-Improved ASPP | 93.97 | 60 | 5.42 | 38.47 |
| YOLOvs5-V-Improved ASPP(No SPP) | 93.65 | 61 | 5.64 | 36.47 |
| YOLOv5s-V-att2-Improved ASPP | 95.18 | 55 | 5.17 | 39.85 |

Combining the analysis in Table 2 and Figure 12, the following conclusions can be drawn:

1. Improvements to the backbone network resulted in a 1.24% improvement in mAP and an increase in the model size of only 2.4 MB. Adding either the attention mechanism or the Improved ASPP module alone, the attention mechanism works better than the Improved ASPP module because the attention mechanism can better assign weights to detection targets. The reason for the relatively small increase in the mAP of the improved ASPP module is that it duplicates the role of the SPP structure in the backbone network. Therefore, we chose to remove the SPP structure for experimental comparison. It can be seen that after removing the SPP module, YOLOv5s-V-att2 (No SPP) decreased by 0.55% compared to the previous mAP. YOLOv5s-V-Improved ASPP (No SPP) decreased by 0.32% compared to the previous mAP. This suggests that the improved ASPP module does duplicate the role of the SPP structure in

the backbone. However, the combined comparison is still the improvement of the attention mechanism that improves the model accuracy more;

2. Both the YOLOv5s-V-improved ASPP and the YOLOv5s-V-att2-improved ASPP have improved on FDR. This represents an improvement in the effectiveness of the improved algorithm in this paper for target detection in complex backgrounds;

3. As Figure 13 shows, the improved model converges faster and has a lower loss value compared to the original model;

4. The three improved ablation experiments resulted in a 3.34% improvement over the original results, with an increase in the model size of only 11.5 M and a reduction in detection speed of 18FPS.

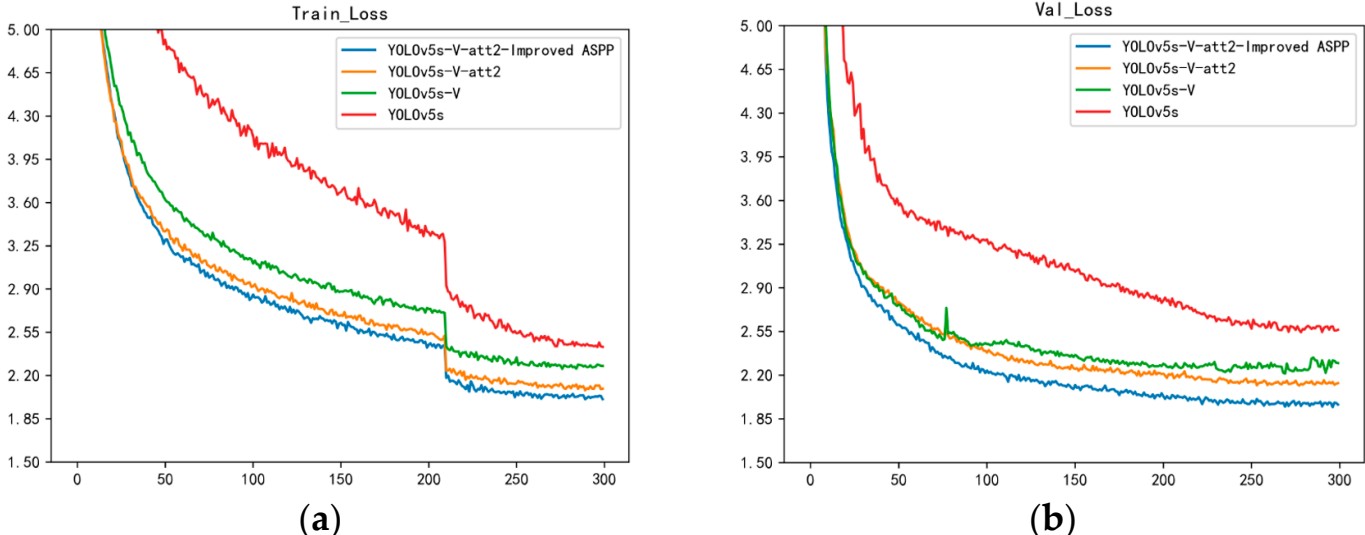

**Figure 13.** (**a**) Training loss curves for the models are shown in Table 4; (**b**) validation loss curves for the models are shown in Table 4.

Table 5 shows the comparison of the detection effect of the test set before and after the model improvement. As Table 5 shows.

To further visualize the attention region in the model, this paper presents the fractional heat map visualization analysis of the predicted results from the fusion model. Table 6 shows the test set image thermal comparison chart. As Table 6 shows: the heat-sensing map covers the majority of the detected target area, with the red being the central area where attention can be continuously spread outwards to reduce.

**Table 5.** Test set image detection results. (**a**) the improved model reduces false detections; (**b**) the improved model enhances the detection of small targets; (**c**,**d**) the improved model enhances the detection of multiple targets in complex backgrounds.

| Original Algorithm | Improved Algorithm |
| --- | --- |

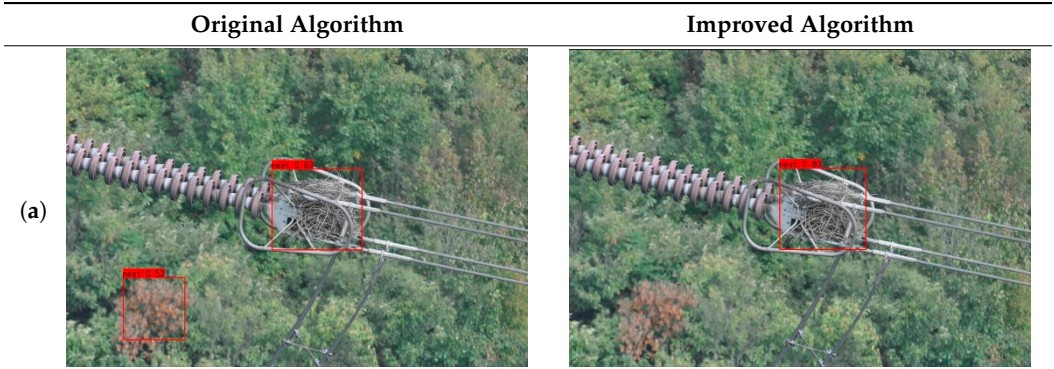

**Table 5.** *Cont.*

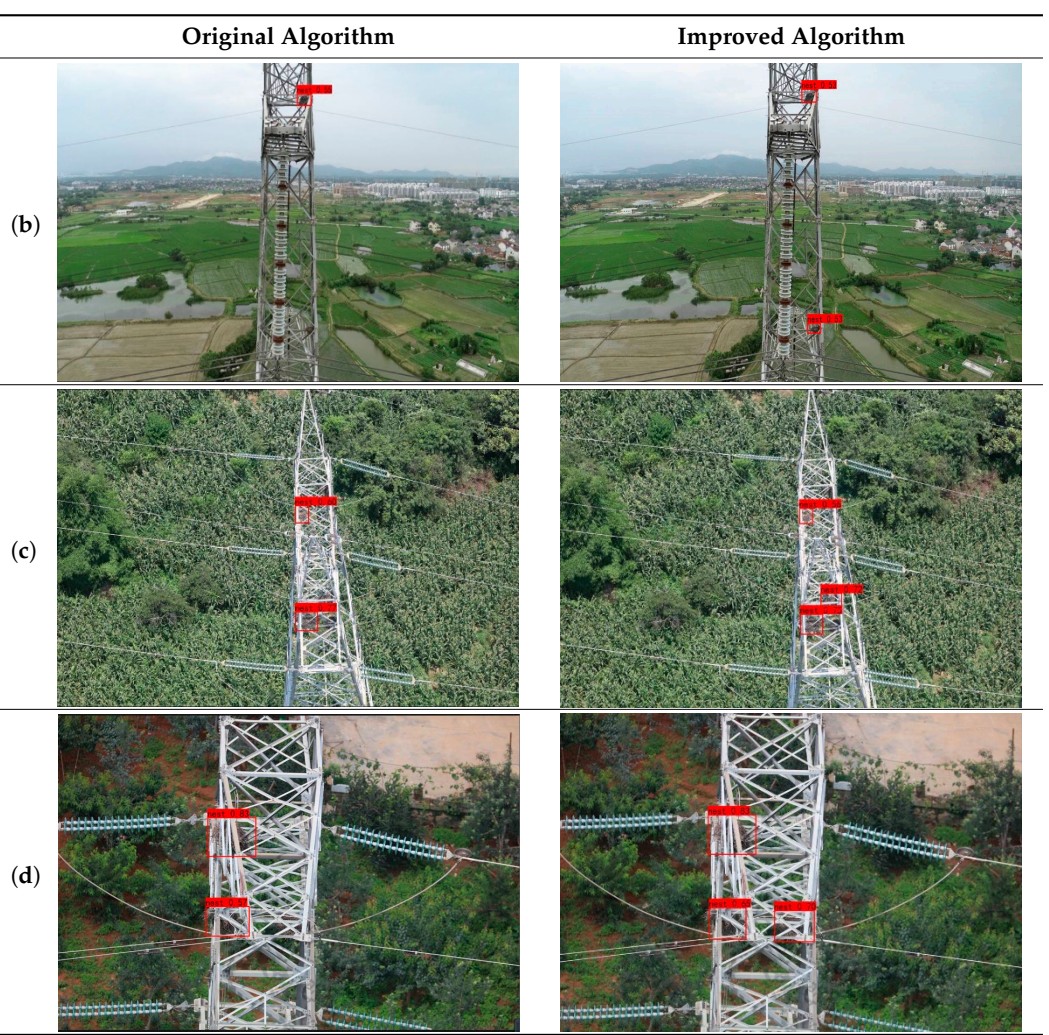

**Table 6.** Test set image thermal comparison chart. (**a**) the improved model detects targets that the initial model sensed but could not detect; (**b**) the improved model provides a significant improvement in the detection of small targets; (**c**) the improved model addresses false detection in complex backgrounds; (**d**) the improved model enhances multi-target detection in complex backgrounds.

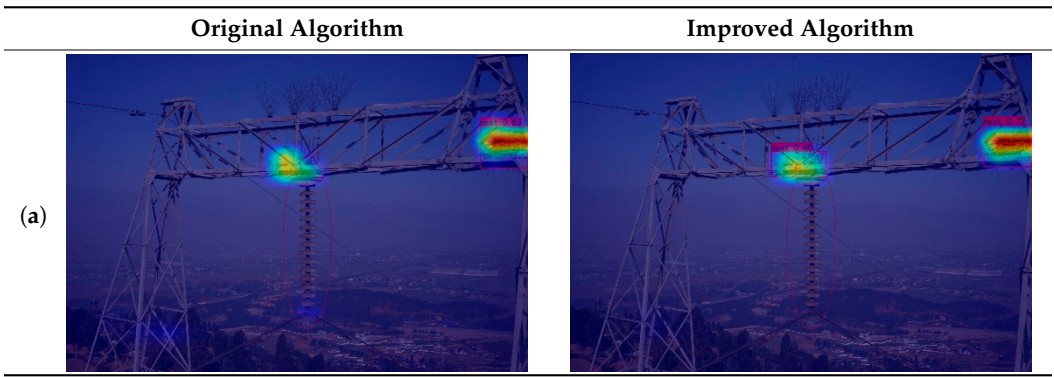

**Table 6.** *Cont.*

| Original Algorithm | Improved Algorithm |
| --- | --- |

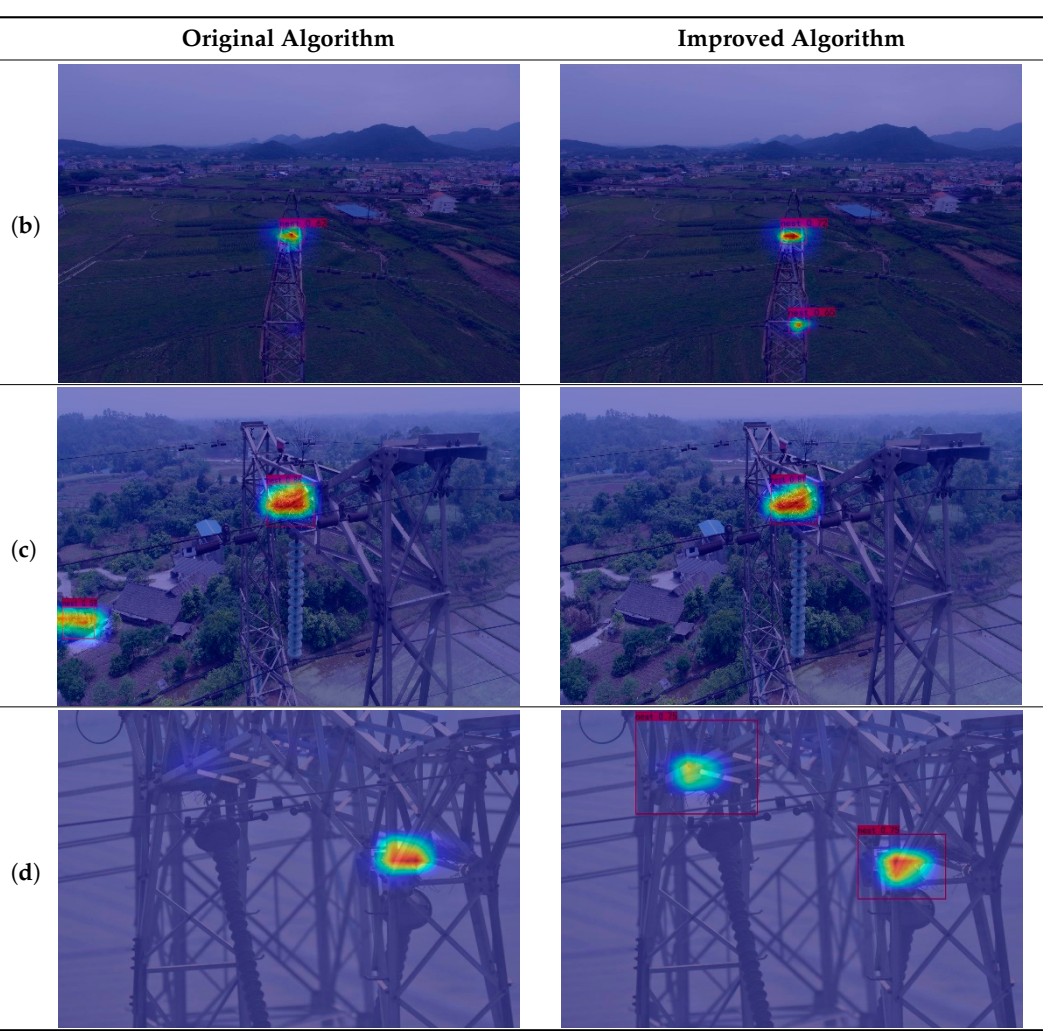

## 5. Embedded Device Deployment

The embedded device port uses the Jetson Xavier NX, which has 384 CUDA cores, 48 Tensor cores, and 2 NVIDIA engines. It can run multiple modern neural networks in parallel, processing high-resolution data from multiple sensors simultaneously. It is suitable for use in drones, portable medical devices, and other systems. The Jetson Xavier NX is shown in Figure 14a, with real-time data acquisition by calling the hardware camera. The acquisition results are shown in Figure 14b, with an image resolution size of 1280 × 720, a detection speed of 10.2 FPS, and a confidence level of 0.82 in the detection of the bird nests.

Table 7 shows the comparison of Jetson Xavier NX test results. As can be seen from Table 7, the improved YOLOv5s model on the Jetson Xavier NX improved detection accuracy by 3.34% over the original YOLOv5s and reduced detection rate by only 0.7 FPS. The actual test showed the conformity of the results of the requirements for real-time detection of bird nests on high-voltage transmission towers during aerial images by drones.

**Table 7.** The comparison of Jetson Xavier NX test results.

| Models | mAP/% | Speed/FPS |
| --- | --- | --- |
| Original YOLOv5s | 91.84 | 10.9 |
| Improved YOLOv5s | 95.18 | 10.2 |

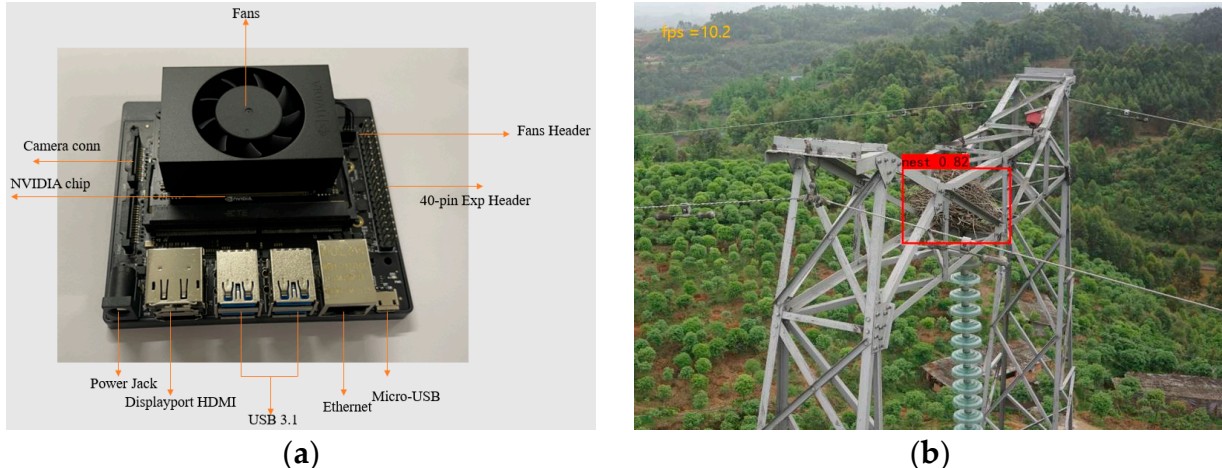

**Figure 14.** (**a**) Structure of Jetson Xavier NX; (**b**) real-time monitoring and collection results.

## 6. Conclusions

In this paper, we improve the algorithm based on YOLOv5s and reconstruct the backbone network of CSPDarkNet by combining the OSA module in VOVNet, which improves the accuracy of the model without increasing the size of the model or the number of parameters computed and facilitates the deployment of the model. Secondly, we investigate the addition of attention mechanisms in different positions of the model to find the most suitable way to add attention mechanisms to improve the detection of small targets. Finally, we investigate the parameters of the ASPP module and select the most suitable parameters for this model to improve the ASPP for multi-target detection in complex backgrounds. The experimental results show that the accuracy of the model improves from 91.84% to 95.18%, the model size is only 39.85 M, an increase of 27.4% over the initial model. The average detection speed is 55 FPS, and the average detection speed, when ported to embedded devices, is 10.2 FPS, which is suitable for deploying embedded edge computer platforms to meet the requirements of real-time detection for UAV inspection.

The algorithm proposed in this paper is currently used to detect bird nests on transmission towers in aerial images and will subsequently investigate the problem of foreign object faults on aerial images of power transmission lines. Eventually, the algorithm is expected to be extended to the detection and monitoring of other major electrical equipment and to enable cloud collaboration, placing some of the target detection functions on the lightweight end.

**Author Contributions:** Conceptualization, G.H. and R.W.; methodology, G.H.; software, R.W.; validation, R.W., Q.Y. and S.L.; formal analysis, G.H.; investigation, L.Z.; resources, M.H., S.Y. and L.Q.; data curation, R.W., Q.Y. and L.Q.; writing—original draft preparation, R.W.; writing—review and editing, G.H., R.W. and L.Q.; visualization, M.H.; supervision, S.Y. and L.Q. All authors have read and agreed to the published version of the manuscript.

**Funding:** This work was supported by the National Key R&D Program of China (No. 2020YFB0905900).

**Institutional Review Board Statement:** Not applicable.

**Informed Consent Statement:** Not applicable.

**Data Availability Statement:** Not applicable.

**Conflicts of Interest:** The authors declare no conflict of interest.

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
