# Peer review of "Detection of Bird Nests on Transmission Towers in Aerial Images Based on Improved YOLOv5s"

_machines, doi:10.3390/machines11020257_

Round 1

Reviewer 1 Report

This paper proposes an improved algorithm based on YOLOV5s for bird nest detection on transmission towers, which is improved by adding OSA module, CBAM module and ASPP module, and the accuracy of the improved model is improved, the work has some reference significance, but this paper mainly has the following problems:

1. The language of the paper should be polished; the paper needs to be revised by a native English speaker. 

2. A detailed explanation should be added for each abbreviation when it first appears.

3. The innovation point of this paper is not clear. Compared with existing methods, what are the biggest differences and advantages?

4. The model size of YOLOv5s in Table 4 is not consistent with Table 2.

5. Conclusion description without increasing the size of the model, but the actual increase, should be reasonably described.

Reviewer 2 Report

Abstract:

1)     Please connect the first two sentences in Abstract. “To further improve the accuracy of model detection of birds' nests on transmission towers 8 in aerial images without significantly increasing the model's size and making it more suitable for 9 edge-end applications.  This paper improves on the lightweight model YOLOv5s.”

Introduction:

1)     Line 32 and 35: There are typos - inefficient455 and detection67. Please correct all similar typos.

2)     Line 77: Please provide the reference for “COCO 77 dataset”

3)     What is conv5 in figure 2? Please provide some description on this.

Recognition of Bird’s nest:

1)     Line 160 : How convolution kernel of size 7 x 7 is selected?

2)     Figure 8: Typo in the caption.

Experimental results and analysis:

1)     Line 217: Please provide reference “the experimental data are sourced from a power company”

2)     Line 222: Who annotated the images? It's always recommended to have a professional bird watcher or wildlife biologist to identify the nest and its species properly.

3)     Please rewrite all sentence that starts with “As can be seen from”

Overall:

The cost of inspecting transmission towers for bird's nests can be quite high due to the need for specialized equipment and personnel. Therefore, the authors should provide a cost analysis and/or comparison of the existing models.

Inclement weather can make it difficult to spot bird's nests on transmission towers, especially if the nests are obscured by foliage or other natural elements. How your model would deal with this problem?

Some bird's nests are built to blend in with their surroundings and can be difficult to spot, even for experienced observers.

Please mention the limitations of the current work and potential future works in a separate section.

Reviewer 3 Report

The main subject of the article is the improvement of the accuracy of a model for the detection of birds' nests on transmission towers in aerial images using a lightweight version of the YOLOv5 algorithm. Most guidelines are respected and well presented except some notes to improve the work:

§  The introduction should include the purpose and/or the main contribution of the current work. The authors likely state what their goal is and how it differs from existing works. They might also mention the specific methods and techniques they use to improve the detection of birds' nests on transmission towers in aerial images, and how these methods differ from those used in previous research. They might also mention the specific results and findings of their work, and how it contributes to the field of bird's nest detection on transmission towers.

§  In the page 12, the author compare the algorithm YOLOv4-Tiny with the other algorithms (Faster R-CNN, SSD, YOLOv3, YOLOv4, and YOLOv5s) in terms of detection accuracy, when in fact, YOLOv4-Tiny is not a mainstream algorithm like the others mentioned and the author might have confused it with YOLOv4. YOLOv4-Tiny is a lightweight version of YOLOv4 and it was not designed to be compared with the other mainstream algorithms in terms of detection accuracy, but rather in terms of detection speed and model size. Try to bring more clarity.

§  In the page 13 the author is suggesting that the attention mechanism works better than the Improved ASPP module alone. However, this conclusion is not well-supported by the data presented, as the text only mentions that the attention mechanism can better assign weights to detection targets and that the Improved ASPP module duplicates the role of the SPP structure in the backbone network. Additionally, the author should have done a comparison between the attention mechanism and the Improved ASPP module together and not alone to have a clear conclusion about which one works better.

§  Include the following references in the list of sources for the article, as they pertain to topics discussed and also mention the deep learning algorithms used.

1. Jabir, B., & Falih, N. (2022). Deep learning-based decision support system for weeds detection in wheat fields. International Journal of Electrical and Computer Engineering, 12(1), 816.

Round 2

Reviewer 2 Report

The authors have responded adequately to all questions and acknowledged the limitations of their study.